# Towards a Formal Characterization of User Simulation Objectives in Conversational Information Access

## ABSTRACT

User simulation is a promising approach for automatically training and evaluating conversational information access agents, enabling the generation of synthetic dialogues and facilitating reproducible experiments at scale. However, the objectives of user simulation for the different uses remain loosely defined, hindering the development of effective simulators. In this work, we formally characterize the distinct objectives for user simulators: training aims to maximize behavioral similarity to real users, while evaluation focuses on the accurate prediction of real-world conversational agent performance. Through an empirical study, we demonstrate that optimizing for one objective does not necessarily lead to improved performance on the other. This finding underscores the need for tailored design considerations depending on the intended use of the simulator. By establishing clear objectives and proposing concrete measures to evaluate user simulators against those objectives, we pave the way for the development of simulators that are specifically tailored to their intended use, ultimately leading to more effective conversational agents.

## CCS CONCEPTS

• **Information systems → Users and interactive retrieval**; • **Computing methodologies → Modeling and simulation**.

## KEYWORDS

User simulation; Conversational information access

**ACM Reference Format:**
Anonymous Author(s). 2024. Towards a Formal Characterization of User Simulation Objectives in Conversational Information Access. In *Proceedings of Proceedings of the 14th International Conference on the Theory of Information Retrieval (ICTIR '24)*. ACM, New York, NY, USA, 9 pages. https://doi.org/10.1145/nnnnnnn.nnnnnnn

## 1 INTRODUCTION

Conversational information access (CIA) has emerged as a new paradigm for information seeking, allowing users to engage in multi-turn dialogues with agents to refine their queries and access relevant information. While significant progress has been made in this area, challenges in training and evaluating CIA agents remain, due in part to the scarcity of large-scale dialogue corpora and the interactive nature of these systems [9, 38]. Indeed, the creation of dialogue corpora involving real users is time-consuming and expensive, and the evaluation of interactive systems presents similar challenges due to the necessity of human participation for accurate assessment. User simulation has been proposed as a potential solution to automate the training and evaluation of CIA agents [3, 12], offering the possibility to generate synthetic dialogues and conduct reproducible experiments at scale, which is notoriously difficult, if not impossible, with real users.

User simulation has been informally defined as the process of mimicking the behavior of real users when interacting with a system in order to accomplish some task [3]. This description is applicable to both training and evaluating CIA agents. However, we argue that each of these uses, i.e., training and evaluation, has distinct objectives. Surprisingly, despite its importance and potential impact, there is limited work on the formal characterization of these objectives in the context of conversational information access. A notable exception is the work by Zhang and Balog [39], who proposed a high-level formalization of requirements for evaluation, based on pairwise comparison of system performance when used with a simulator. However, this approach has limitations, as it does not account for absolute differences in evaluation scores, potentially masking substantial variations in system performance. For training, we are not aware of any work on formalizing objectives from a user simulation perspective, despite the widespread use of simulators.

The current paper aims to fill this gap by formally characterizing the distinct objectives for different uses of user simulation in conversational information access, and by proposing corresponding metrics for measuring the extent to which these objectives are met. For training, we posit that the objective is to closely mirror the behavior of real users. This can be expressed in terms of the similarity of dialogue policies, which determine the next user action given the current context. In contrast, for evaluation, the objective shifts to accurately estimating the performance of real users when interacting with a CIA agent in order to accomplish some task. Given the apparent alignment of these objectives, it is natural to ask the question whether optimizing for one objective would inherently lead to improvements on the other objective. Our initial empirical study reveals that this is not necessarily the case. We find several instances where simulator A outperforms simulator B on the training objective, but it is the other way around for the evaluation objective. This suggest the need for distinct design considerations when developing user simulators for training versus evaluating systems.

In summary, the main contributions of this work are twofold (1) a formalization of the problems related to the two uses of simulation in CIA, training and evaluation, identifying their distinct objectives and measures for evaluating simulators against those objectives, and (2) an empirical study showing that the two objectives are not necessarily aligned. This work thus lays the foundation for the development of more effective, purpose-built user simulators, ultimately leading to more user-centric conversational agents.

*ICTIR '24, ,*
© 2024 Association for Computing Machinery.
ACM ISBN 978-1-4503-XXXX-X/18/06...$15.00
https://doi.org/10.1145/nnnnnnn.nnnnnnn

## 2 RELATED WORK

Introduced in the fields of information retrieval in the 1970s [7] and dialogue systems a few decades later [10], user simulation approaches have recently gained renewed attention due to their potential for training and evaluating conversational information access (CIA) agents [1, 2].

### 2.1 User Simulation for Training

The objective of training a CIA agent is to learn a dialogue policy that selects the best action to take given the current dialogue state [30]. Several approaches have been proposed to tackle this problem including deep learning using human-human [36], synthetic, or hybrid corpora [13] as well as reinforcement learning (RL) [17]. User simulators can be used to generate a synthetic corpus for offline training, or mimic a human a dialogue participant in a reinforcement learning setting. In the first case, user simulation is a fast and cheap way to generate large amounts of data and to help overcome the problem of data sparsity. Indeed, the collection of dialogues involving humans is cumbersome and expensive. The Simulated Agenda Dataset [23] is an example of a synthetic corpus built with the help of user simulators. While the creation of synthetic corpora is an interesting question, it is not the focus of this work. In the case of RL, the decision-making process of selecting the next action is commonly modeled as a Markov Decision Process (MDP) [17]. In this context, user simulation allows for dynamic interactions with the conversational agent. Furthermore, it allows for an exhaustive exploration of the state-action space, which is otherwise limited when using a fixed corpus [30]. Schatzmann et al. [29] train a policy using a user simulator that does not require training data. Conversely, Lin et al. [20] perform a comparison of dialogue policies trained with the PPO algorithm [31] using different user simulators. The approach of training a user simulator first and then using it in a RL setting is often used in the field [17].

### 2.2 User Simulation for Evaluation

The evaluation of CIA agents is an open challenge and is thus an active area of research [3, 38]. None of the existing evaluation methodologies (offline, online, and user studies) enables the comparison of multiple CIA systems using reproducible experiments. Offline test collections are static in nature and fail to capture the interactive nature of conversations, while there is an inherent lack of reproducibility, coupled with challenges of performing evaluation at scale, when real users are involved. User simulation is presented as a promising solution to complete existing evaluation methodologies [1]. Indeed, with a user simulator it is possible to quickly and inexpensively perform evaluation based on a large number of dialogues. Moreover, simulated users can be controlled, thereby enabling reproducible experiments. In the literature, user simulation has been used to evaluate CIA agents at different granularity levels. For example, Zhang and Balog [39] performed an evaluation at the conversation level, while others [24, 28, 32] evaluated the mixed-initiative abilities of CIA agents at the turn level.

Evaluation metrics may be divided into two categories: single-turn and multi-turn metrics (also referred as end-to-end metrics) [26]. One major limitation of single-turn metrics is that they do not consider the dialogue structure, which is an important aspect of CIA and can impact the overall user experience [38]. However, they are useful when a specific feature of a CIA agent is in focus. For example, Sekulić et al. [32] used BLEU [25] and ROUGE [18] to automatically assess the quality of their simulator's generated utterance; these particular metrics have limitations when applied to dialogues, yet they are widely used due to the lack of better alternatives [22]. In the case of conversational search, previous work [24, 28] have used information retrieval metrics such as normalized discounted cumulative gain and mean reciprocal rank to evaluate the relevance of retrieved items. Conversely, multi-turn metrics take into consideration multiple dialogue turns or entire dialogues. These metrics commonly evaluate the conversational agent's ability to achieve a given goal. Examples include the broadly used success rate and the expected conversation satisfaction metrics [21]. While they are insightful, they do not consider the quality of the dialogue in terms of structure, naturalness, and coherence. Out of these aspects, naturalness has attracted the most attention and metrics such as D-BLEU [14] and SUPER [27] were proposed to assess it.

### 2.3 Requirements for User Simulators

Even though there is a large body of work leveraging user simulation for training and evaluation of CIA agents, the work by Zhang and Balog [39] is the only one, to the best of our knowledge, that mathematically formalizes the requirements for a user simulator for the purpose of system evaluation. We believe that such formalization is important to identify the requirements and desiderata for user simulation in CIA and provide a solution to verify if they are met. Balog and Zhai [3] also argue that interpretability of user simulators is an important requirement for evaluation as it allows for a comprehensive understanding of the results. On the other hand, interpretability is not critical for training. Additional work have discussed requirements and desiderata for user simulation [1, 5, 24], however, these are articulated in qualitative terms, lacking quantifiable measures for assessing their fulfillment.

## 3 PROBLEM STATEMENT

We formally define the two main uses of user simulation in conversational information access: training and evaluation. This formalization aims to make the objectives for each use of user simulation explicit, thereby providing a framework that (i) enables a comparative analysis of different uses of simulation with regards to objectives and (ii) guides the design of appropriate measures for evaluating simulators. We emphasize the importance of studying user simulation not in isolation, but in relation to how it interacts with the conversational agent within the specific use context. In this section, we start in Section 3.1 by defining fundamental concepts. Then, we propose a formalization for training and evaluation in Sections 3.2 and 3.3, respectively.

### 3.1 Preliminaries

We define key concepts used in the remainder of this work. For ease of reference, their notation and description are included in Table 1.

We consider a task-based dialogue setting, where a user $u$ interacts with a conversational agent $CA$ for the purpose of completing

**Table 1: Notation.**

| Symbol | Description |
|--------|-------------|
| $CA$ | Conversational agent |
| $u$ | User |
| $U$ | Set of users (i.e., user population) |
| $U^*$ | Simulated users |
| $G$ | Goal to complete |
| $d$ | Dialogue |
| $D$ | Set of dialogues |
| $\mathcal{A}$ | Action space |
| $\mathcal{S}$ | Dialogue state space |
| $t$ | Dialogue turn |
| $x$ | An utterance |
| $\pi_\theta$ | Dialogue policy parametrized by $\theta$. It determines the next interaction |
| $m(d, G)$ | Measure of the completion of goal $G$ in dialogue $d$ |
| $M(CA, U)$ | Measure of the performance of a conversational agent $CA$ when used by a set of users $U$ |
| sim | Similarity function |

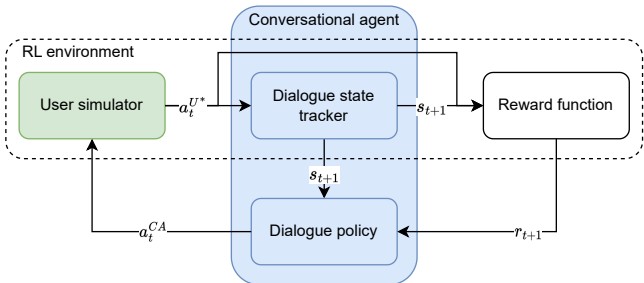

**Figure 1: Place of user simulator in the training process.**

some goal $G$. For example, the goal may be to learn about astrophysics or to find a new laptop under \$700. We assume that the completion of the goal may be measured by some metric $m$ to assess the performance of $CA$. In the course of the dialogue, participants take turns $t$ in issuing *utterances* $x$: $d = [x_0^{CA}, x_0^u, x_1^{CA}, x_1^u, \ldots, x_n^{CA}, x_n^u]$.

With user simulation, the aim is to create a simulated user $u^*$ that can substitute the real user $u$ in the dialogue with $CA$. In practice, we are typically interested in a set of users $U$ that are representative of the expected user base of the system, rather than relying on a single individual. Thus, we let $U^*$ denote a user simulator that is meant to mimic the behavior of the user population $U$. We assume that each user is characterized by a set of properties, such as personality traits and preferences. (Some properties might be defined on the population level, while others are specific to each individual.) The user simulator $U^*$ then serves as model predicting the actions of a particular user $u^*$ with specific characteristics in a given context.

## 3.2 Training Formalization

Training a conversational agent consists in learning a *dialogue policy* that dictates how it should respond in a given dialogue situation. Without loss of generality, we can define the policy as a mapping from a set of possible states $\mathcal{S}$ to a set of possible actions $\mathcal{A}$, $\pi_{CA} : \mathcal{S} \rightarrow \mathcal{A}$. The *state* $s \in \mathcal{S}$ may take different forms, such as an explicit knowledge structure, where the user goal is represented as a set of slots and values (as in frame-based dialogue systems [15]), or as a latent representation in an embedding space (see, e.g., [11, 19]). *Actions* $a \in \mathcal{A}$ represent the intent behind the response, which is ultimately given as a natural language utterance $x$. In traditional state-based dialogue architectures, there is a separate natural language generation component that creates the utterance from the action [11, 15, 39]. More end-to-end simulator architectures might generate utterances jointly with the underlying actions [19], or generate only the natural language utterance [8, 16].

In the latter case, we assume that the action can be inferred from the utterance.

The mapping defined by the policy may be deterministic, corresponding to a rule-based system, or stochastic and adaptive, which is typical for probabilistic policies that may be defined based on empirical observations or be automatically learned. We focus on this latter category of learned policies, which allows for more flexible and context-sensitive responses [6]. Specifically, reinforcement learning is often employed as a learning paradigm, where the conversational agent continuously refines its policy through trial-and-error based on feedback from user interactions; see Fig. 1. The task is expressed as an optimization problem, based on the notion of *reward*. The reward function $R$ assigns a numerical value to transitions from one state to another due to an action: $R : \mathcal{S} \times \mathcal{A} \times \mathcal{S} \rightarrow \mathbb{R}$. The goal then is to find an optimal policy $\pi_{CA}^*$ that maximizes the expected cumulative reward:

$$\pi_{CA}^* = \max_{\pi_{CA}} \mathbb{E}\Big[ \sum_{t=0}^{\infty} \gamma^t R(s_t, a_t, s_{t+1}) | \pi_{CA} \Big], \tag{1}$$

where $\gamma$ is a discount factor ($0 \leq \gamma < 1$) that determines the present value of future rewards.

Notice that the formulation in Eq. (1) does not explicitly consider the user simulator as part of the learning process. Since the conversational agent and the user interact to achieve a shared goal through a series of utterances, the actions taken by the user play a pivotal role in determining the subsequent state of the conversational agent ($s_{t+1}$). Consequently, the reward function may be redefined such that it considers the action space of the conversational agent ($\mathcal{A}_{CA}$) as well as the action space of the user ($\mathcal{A}_U$), when making state transitions $R : \mathcal{S} \times \mathcal{A}_{CA} \times \mathcal{A}_U \times \mathcal{S} \rightarrow \mathbb{R}$. The objective for the optimal policy may then be written as:

$$\pi_{CA}^* = \max_{\pi_{CA}} \mathbb{E}\Big[ \sum_{t=0}^{\infty} \gamma^t R(s_t, a_t^{CA}, a_t^u, s_{t+1}) | \pi_{CA}, \pi_U \Big]. \tag{2}$$

With the role of the user policy $\pi_U$ made explicit in Eq. (2), we now turn to the examination of objectives for effectively simulating the policy of real users.

*3.2.1 Simulation Objectives.* In order to facilitate the learning of the dialogue policy of a conversational agent, simulated users should act the same way as real users would act in a given dialogue situation. Formally, the similarity of the simulated user policy $\pi_{U^*}$ and the real user policy $\pi_U$ need to be maximized with regards

to some similarity function:

$$\pi_{U^*}^* = \max_{\pi_{U^*}} sim_\pi(\pi_{U^*}, \pi_U) \,. \tag{3}$$

Given the above, a relative ranking of user simulators for the training objective may be established based on their similarity to the real user policy: $sim_\pi(\pi_{U_1^*}, \pi_U) > sim_\pi(\pi_{U_2^*}, \pi_U) \implies U_1^* > U_2^*$.

A significant challenge with this formulation lies in the fact that $\pi_U$ is not directly observable. This is because we lack direct insight into the thoughts of real users during their interactions with the conversational agent. Hence, a proxy is needed for estimating the similarity between $\pi_U$ and $\pi_{U^*}$.

*3.2.2 Evaluating Simulation.* As $\pi$ defines the conversational strategy, it can be indirectly observed through the generated dialogues. We thus consider historical dialogues with real users ($D_U$) and with simulated users ($D_{U^*}$) as proxies of the respective policies:

$$sim_\pi(\pi_U, \pi_{U^*}) \approx sim_D(D_U, D_{U^*}) \,, \tag{4}$$

where $sim_D$ is a similarity function for comparing dialogues. Representing dialogues as a sequence of actions ($d = [a_0^{CA}, a_0^u, a_1^{CA}, a_1^u, \ldots, a_n^{CA}, a_n^u]$), similarity may be defined on the level of entire conversations or on the level of individual turns.

Specifically, *conversation-level* similarity measures could adapt methods from the fields of automatic summarization and machine translation, such as ROUGE-L and METEOR. The idea is that a dialogue may be seen as a sequence of words, where the words are actions. Then, the aforementioned measures can assess the extent with which simulated users exhibit the same behavioral patterns as reals user by comparing the subsequences of actions. Note that the notion of ordering is important and should be taken into consideration by the chosen similarity measure. Commonly, the measures from automatic summarization and translation compare the generated text against a set of reference translations/summaries. However, in our case, we want to compare two sets of dialogues ($D_U$ and $D_{U^*}$) which do not have pairwise correspondence. Thus, a possible adaptation is to compute the measure for all possible pairs of dialogues and then aggregate the results:

$$sim_D(D_U, D_{U^*}) = aggr(sim_d(d, d^*) \mid d \in D_U, d^* \in D_{U^*}) \,, \tag{5}$$

where $sim_d$ is the similarity measure between two dialogues, e.g., ROUGE or METEOR, and *aggr* is an aggregation function, e.g., average, median, or maximum. For *turn-level* similarity measures, we can characterize each system action in terms of the distribution of user actions that follow that action, i.e., $P_\pi(a^u|a^{CA})$ where $a^u \in \mathcal{A}_U$ and $a^{CA} \in \mathcal{A}_{CA}$. This distribution could correspond to the empirical distribution from dialogues or be estimated based on heuristics and intuition. The distributions $P_{\pi_U}$ and $P_{\pi_{U^*}}$ may be compared using similarity measures of probability distributions, such as Jensen-Shannon divergence. Unlike conversation-level measures, the notion of ordering is lost here. Indeed, the distribution of user actions is independent of the conversational context except for the current system action. Note that one can consider using both *conversation-* and *turn-level* metrics to get a more comprehensive assessment of the similarity between the user policies. Indeed, as they take into account either the global or local context of a dialogue, they capture different aspects of user behavior, such as overall strategy versus response to a specific system action.

## 3.3 Evaluation Formalization

User simulation allows for a time- and cost-efficient evaluation of conversational agents by reducing the reliance on real users, thereby enabling automatic evaluation at scale [3]. However, the validity of the user simulator is critical to ensure the trustworthiness and reliability of the findings. The objective of evaluation using user simulation has previously been defined by Zhang and Balog [39] as: "an automatic assessment of the agent such that it is predictive of its performance with real users." In order to verify this, they propose to compare the performance of two conversational agents $CA_1$ and $CA_2$ when used with $U$ and $U^*$, and formulate the following condition to be satisfied: $M(CA_1, U) < M(CA_2, U) \implies M(CA_1, U^*) < M(CA_2, U^*)$, where $M(CA, U)$ represents some measure of performance when $CA$ is used by a set of users $U$.

A main issue with the above pairwise comparison that it does not take the absolute performance differences into account. As such, it cannot be used to establish a relative ranking of user simulators. For example, let us assume that we obtain the performance measurements for two conversational agents with real users $M(CA_1, U) = 0.1$ and $M(CA_2, U) = 0.2$. Let one user simulator $U_1^*$ estimate the performances to be $M(CA_1, U_1^*) = 0.05$ and $M(CA_2, U_1^*) = 0.8$, while another user simulator $U_2^*$ predicts these to be $M(CA_1, U_2^*) = 0.15$ and $M(CA_2, U_2^*) = 0.25$. While both $U_1^*$ and $U_2^*$ agree in how they rank $CA_1$ and $CA_2$ relative to each other, which is in agreement with what we got from real users, intuitively, $U_2^*$ is a better and more useful simulator as it produces estimates that are closer to the real performance measures in absolute terms. Based on this observation, we propose an adaption of the requirements from [39]—one that facilitates the comparison between user simulators with regards to the system evaluation objective.

*3.3.1 Simulation Objectives.* In order to automatically evaluate conversational agents, the performance measures obtained from simulated users should closely approximate those obtained from real users, that is:

$$M(CA, U) \simeq M(CA, U^*) \,. \tag{6}$$

This notion of near equivalence, however, is challenging to operationalize directly; therefore, we introduce a threshold parameter $\varepsilon$ to define acceptable levels of approximation with regards to a given evaluation metric:

$$|M(CA, U) - M(CA, U^*)| \le \varepsilon \,. \tag{7}$$

Depending on the use case and aim for evaluation, $\varepsilon$ can be set to different values, hence, quantifying the notion of a "good enough" user simulator [2].

The relative ranking of user simulators for the evaluation objective can be established by comparing their performances to that of real users: $|M(CA, U) - M(CA, U_1^*)| < |M(CA, U) - M(CA, U_2^*)| \implies U_1^* > U_2^*$.

*3.3.2 Evaluating Simulation.* Now that we have defined the objectives, we focus on how to assess the performance of the conversational agent for a given set of users. Here, we focus on measuring performance with regards to the agent's ability to aid users in completing some goal $G$, based on a set of dialogues $D_U$ between the

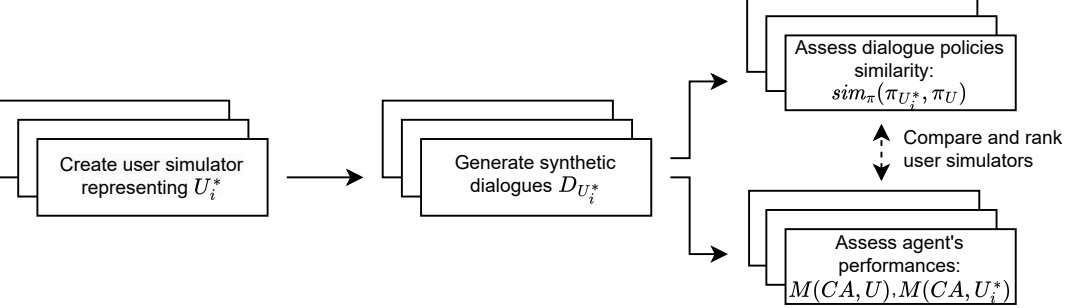

Figure 2: Overview of our methodology. The user simulators are employed to generate synthetic dialogues that are used for the assessment of simulators for the training objective (dialogue policy similarity) and evaluation objective (agent performance). The dashed line represents the comparison between the user simulators for the two objectives.

agent and users:

$$M(CA, U) = \frac{1}{|D_U|} \sum_{d \in D_U} m(d, G) , \qquad (8)$$

where $m(d, G)$ is a metric that assesses if the goal $G$ was completed or not. This assessment can be binary, e.g., completed or not, or graded, corresponding to the degree of completion. While this formulation implies that $m$ is a *conversation-level* metric, one can choose to use a *turn-level* metric $m(t, G)$, such as recall in conversational search [32], if it is better suited for their use case. In that case, Eq. (8) should be adapted to iterate over the turns $t$ of all the dialogues in $D_U$.

Additionally, it is possible to use statistical significance testing to evaluate the objectives with a certain confidence level. Let $P_{M(CA,U)}$ and $P_{M(CA,U^*)}$ be the distribution of performances for real and simulated users, $U$ and $U^*$, respectively (based on the dialogue-level performances $m(d, G)$ that are averaged in Eq. (8)). Taking the formulation in Eq. (6), we can extract the null hypothesis $H_0 = P_{M(CA,U)} \not\simeq P_{M(CA,U^*)}$ and the alternative hypothesis $H_1 = P_{M(CA,U)} \simeq P_{M(CA,U^*)}$. Then, a statistical test can be employed to determine whether the requirement is satisfied by either rejecting or not rejecting the null hypothesis $H_0$, with a specified level of confidence.

## 4 ANALYSIS

In this section, we study the relationships between the objectives for training and evaluation. More specifically, we are interested in the potential implications between these objectives. Consequently, we seek to answer the following research question: *If user simulator A outperforms user simulator B on the training objective, does this imply that A will also outperform B on the evaluation objective, and vice versa?* We start in Section 4.1 by describing the methodology applied. Then, we introduce the user simulator used in our experiments (Section 4.2), followed by our experimental setup (Section 4.3). Finally, we present the results and discuss the findings with regards to our research question (Section 4.4).

### 4.1 Methodology

The methodology considers $N$ conversational agents and corresponding user populations. For each conversational agent $CA$, we perform the following steps (Fig. 2):

(1) Create various user simulators $U_i^*$ to be compared with regards to the training and evaluation objectives.
(2) Generate synthetic dialogues ($D_{U_i^*}$) between the reference conversational agent $CA$ and each user simulator $U_i^*$. The dialogues are generated by sampling the actions of each participant according to their dialogue policy. Moreover, they have an associated (binary) success outcome: successful or not.
(3) Compute the different metrics associated with the training and evaluation objectives. For training, we consider two metrics to measure the similarity between the reference and simulated user populations at the turn- and conversation-levels. The turn-level similarity is computed with the Jensen-Shannon divergence (JSD) between the policies of the user simulator and the reference user population. The conversation-level similarity is an aggregation of the ROUGE-L scores between the synthetic dialogues $D_{U_i^*}$ and the reference dialogues $D_U$. For evaluation, we use the success rate to assess the performance of the conversational agent.
(4) Establish a relative ordering of user simulators $U_i^*$ based on the computed metrics.

### 4.2 User Simulator

For our experiments, we employ a user simulator that is characterized by three elements: (1) user model, (2) interaction model, and (3) outcome prediction.

*User model.* The user model defines the characteristics of simulated users with respect to patience and inclination. Patience is associated with the time a user is willing to be engaged to complete the task. Inclination refers to the overall attitude of the user towards goal completion; higher inclination increases the likelihood of a successful dialogue outcome.

**Table 2: Datasets description.**

| Dataset | # Dialogues | Avg. dialogue length | Domain | Success label |
|---|---|---|---|---|
| DSTC1 [37] | 15,577 | 24.217 ± 22.145 | Bus timetable | × |
| DSTC2 [37] | 2,117 | 10.171 ± 4.497 | Restaurant | × |
| ODE[a] | 25 | 15 ± 8.679 | Dataset exploration | ✓ |
| SCS [34] | 38 | 1.579 ± 0.5 | Open search | × |
| MG-ShopDial [4] | 63 | 20.15 ± 9.474 | E-commerce | × |

[a] https://github.com/svakulenk0/ODExploration_data

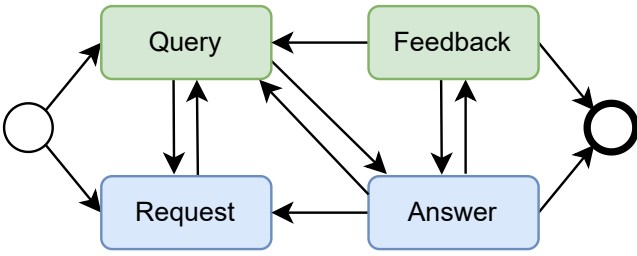

**Figure 3: QRFA model [35]. User and agent actions are shown in green and blue, respectively.**

**Table 3: User model used to parametrize user simulators with respect to patience (time a user is willing to engage) and inclination (attitude towards goal completion).**

| User sim. | Personality | Patience | Inclination |
|---|---|---|---|
| $U_1^*$ | Impatient and critical | -0.9 | -0.9 |
| $U_2^*$ | Impatient and positive | -0.9 | 0.9 |
| $U_3^*$ | Patient and critical | 0.9 | -0.9 |
| $U_4^*$ | Patient and positive | 0.9 | 0.9 |
| $U_5^*$ | Neutral | 1e-5 | 1e-5 |

*Interaction model.* The interaction model is based on the QRFA (query, request, feedback, and answer) model [35]. The model defines two user actions, i.e., query and feedback, and agent actions, i.e., request and answer; see Fig. 3. It has the advantage of being simple and generalizable to a wide range of applications. Using this model, we define transition probabilities between the agent and user actions, that is, each agent action conditions the next user action following a probability distribution.

*Outcome prediction.* The outcome is predicted by a logistic regression model $f$ that assigns a binary outcome to a dialogue, i.e., successful or not.[1] Our model uses five input features: dialogue length $l$, last action from the user $a_u^t$ and from the conversational agent $a_{CA}^t$, patience $p$ and inclination $i$ of the user. We chose this particular combination of features as they capture the notion of effectiveness of the conversational agent (with $l$, $a_u^t$, and $a_{CA}^t$), while also considering the user's perspective (with $p$ and $i$). Formally, we define $f$ based on the sigmoid function:

$$f = \frac{1}{1 + \exp(-h(l, a_u^t, a_{CA}^t, p, i))} \quad (9)$$

where $h$ computes a score based on input features. It can be a simple linear model or a more complex model, e.g., a neural network. For this analysis, we define $h$ as a linear model:

$$h(l, a_u^t, a_{CA}^t, p, i) = w_1 \frac{p}{l} + w_2 \tanh(i) \mathbb{1}(a_u^t = F) + w_3 \mathbb{1}(a_{CA}^t = A),$$
$$(10)$$

where $w_1$, $w_2$, and $w_3$ are feature weights, and $\mathbb{1}$ is the indicator function. Patience and dialogue length go hand in hand as they are related to the time a user is engaged in the dialogue. Therefore, we combine them in the first term of the model to account for this correlation. Within the QRFA interaction model, the feedback action

[1] We note that other models, such as support vector machine or Bayesian network, could also be used instead.

does not distinguish between positive and negative feedback, hence, we assume that $i$ can be used to polarize the feedback action. In this analysis, we use a hyperbolic tangent function for the polarization, noting that other functions (e.g., sigmoid) could also be used.

For the analysis, we set $w_1$ and $w_2$ to 1, and $w_3$ to 0.5 inspired by the empirical study by Siro et al. [33]. However, we acknowledge that the choice of the weights is somewhat arbitrary and may not be optimal with regards to real-word scenarios.

### 4.3 Experimental Setup

For the experiments, we consider five conversational agents $\{CA_1, CA_2, \ldots, CA_5\}$ with corresponding user populations, and five user simulators $\{U_1^*, U_2^*, \ldots, U_5^*\}$ parametrized to simulate these user populations. These are associated with five datasets that have each been annotated according to the QRFA interaction model; see Table 2. These datasets allow us to ground our analysis in real data, hence, integrating the notion of realistic user behavior. Furthermore, we can assume that each dataset reflects different user behavior regarding the completion of an information-seeking task. Indeed, the datasets are from different domains, e.g., restaurant and open-domain search, and have different dialogue lengths indicating that users employ different strategies to complete their tasks.

Using the QRFA annotations, we can extract the transition probabilities between the actions for both the user simulator and conversational agent from each dataset. The transition probabilities are used as a proxy for the dialogue policy. Note that some dialogues in the datasets are not annotated with a success label that is required to assess the success rate, hence, the evaluation objective. Then, we characterize each user simulator with regards to patience and inclination (Table 3). For the sake of simplicity, we consider that each user within the simulated user population is associated with the

**Table 4: Comparison of user simulators with respect to the training and evaluation objectives. User simulators are ranked from the best to the worst (left to right) based on the computed metrics, i.e., Jensen-Shannon divergence (JSD) and aggregated ROUGE-L for training, and the absolute difference in success rate for evaluation. (Note that for JSD lower values correspond to better performance.)**

| Reference | Training objective | | Evaluation objective |
|---|---|---|---|
| $CA_1$ | JSD: | $U_2^*$ (0.211) > $U_3^*$ (0.357) > $U_5^*$ (0.498) > $U_4^*$ (0.543) | $U_4^* = U_5^*$ (0.534) > $U_2^*$ (0.382) > $U_3^*$ (0.132) |
| | ROUGE-L: | $U_3^*$ (0.495) > $U_2^*$ (0.493) > $U_5^*$ (0.492) > $U_4^*$ (0.429) | |
| $CA_2$ | JSD: | $U_1^*$ (0.211) > $U_4^*$ (0.383) > $U_3^*$ (0.412) > $U_5^*$ (0.494) | $U_1^*$ (0.262) > $U_4^* = U_5^*$ (0.046) > $U_3^*$ (0.014) |
| | ROUGE-L: | $U_3^*$ (0.52) > $U_1^*$ (0.456) > $U_4^*$ (0.457) > $U_5^*$ (0.405) | |
| $CA_3$ | JSD: | $U_4^*$ (0.330) > $U_1^*$ (0.357) > $U_2^*$ (0.412) > $U_5^*$ (0.520) | $U_1^*$ (0.778) > $U_2^*$ (0.166) > $U_4^* = U_5^*$ (0.08) |
| | ROUGE-L: | $U_1^*$ (0.525) > $U_4^*$ (0.414) > $U_2^*$ (0.5) > $U_5^*$ (0.458) | |
| $CA_4$ | JSD: | $U_3^*$ (0.330) > $U_2^*$ (0.383) > $U_1^*$ (0.543) > $U_5^*$ (0.554) | $U_3^*$ (0.678) > $U_1^*$ (0.666) > $U_2^*$ (0.034) > $U_5^*$ (0) |
| | ROUGE-L: | $U_1^*$ (0.513) > $U_2^*$ (0.508) > $U_5^*$ (0.503) > $U_3^*$ (0.462) | |
| $CA_5$ | JSD: | $U_2^*$ (0.494) > $U_1^*$ (0.498) > $U_3^*$ (0.520) > $U_4^*$ (0.554) | $U_1^*$ (0.312) > $U_3^*$ (0.164) > $U_2^*$ (0.068) > $U_4^*$ (0) |
| | ROUGE-L: | $U_1^*$ (0.55) > $U_3^*$ (0.515) > $U_2^*$ (0.454) > $U_4^*$ (0.311) | |

**Table 5: Pairwise Kendall's $\tau$ correlation between the metrics for the training and evaluation objectives.**

| Reference | JSD, ROUGE-L | JSD, evaluation | ROUGE-L, evaluation |
|---|---|---|---|
| $CA_1$ | 0.667 | 0.333 | 0.667 |
| $CA_2$ | 0.333 | -0.333 | 0.333 |
| $CA_3$ | 0.667 | -0.333 | -0.667 |
| $CA_4$ | -0.333 | -0.667 | 0 |
| $CA_5$ | 0.333 | -0.333 | -1 |

same personality.[2] The values for $p$ and $i$ are bounded in the range $[-1, 1]$ and chosen to reflect extreme cases (e.g., very impatient or very positive), in addition to a neutral case (i.e., $U_5$). Based on these values and the dialogues, we enrich the datasets with success labels. The labels are assigned with the function $f$ defined in Section 4.2.

Our experiments are inspired by the leave-one-out cross validation approach: we consider one conversational agent and its corresponding user population as reference, and compare the four user simulators against this reference. (For example, considering $CA_1$ as the reference, the user simulators compared are $U_2^*, U_3^*, U_4^*$, and $U_5^*$.) The number of synthetic dialogues generated for each simulated user population is set to 500 (i.e., second step of the methodology). We repeat this process for all possible reference conversational agents. The results are reported in Table 4.[3]

### 4.4 Results

Our interest lies in the comparison of the user simulators with respect to the training and evaluation objectives in order to determine if optimising for one objective would also lead to improvements on the other objective. From the results presented in Table 4, we make two main observations. First, focusing on the training objective alone, we see that the turn- and conversation-level evaluation measures (JSD and aggregated ROUGE-L) do not agree on which simulator is ranked first. To quantify the agreement in the rankings,

we compute the Kendall's $\tau$ correlation between JSD and ROUGE-L, see Table 5, and find that it varies between -0.333 and 0.667, indicating that the metrics are not strongly correlated. This supports the idea that the level of context considered by the metrics provides different insights into the behavior of the simulated user populations. It also highlights that the choice of the similarity metric is an important design decision and also suggests that further research is needed on measures that can integrate local and global behavior.

Second, comparing simulators on the training vs. evaluation objectives, we observe several cases where there is a disagreement in the relative ordering for a pair of user simulators. For example, for the reference conversational agent $CA_1$, we find $U_2^* > U_4^*$ for the training objective (according to both JSD and aggregated ROUGE-L), while for the evaluation object (success rate) it is $U_4^* > U_2^*$. In Table 5, we report the pairwise Kendall's $\tau$ correlation between the metrics for the training and evaluation objectives. Interestingly, we see that Jensen-Shannon divergence and the absolute difference of success rate tend to disagree on the rank of a user simulator ($\tau$ is negative in 80% of the cases studied). In case of the ROUGE-L measure, we do not observe a clear trend for $\tau$, suggesting that additional experiments are needed to understand its correlation with the evaluation metric. Overall, our results provide evidence that training and evaluation objectives are not always aligned, hence, designing/selecting a user simulator should consider its use, i.e., training vs. evaluation.

---

[2]This choice was made in the interest of simplicity. A straightforward extension to simulating variations in personalities within a user population would be to characterize the personality traits, i.e., patience and inclination, as distributions.

[3]The code and data to run these experiments are made available at: https://anonymous.4open.science/r/usersim-objectives-D177/

# 5 DISCUSSION

We acknowledge that the high-level requirements/desiderata of *interpretability* and *realism* have been identified in the literature and are considered important for a widespread adoption of user simulation [2, 3]. Below, we discuss how our proposed objectives related to these. Interpretability is an objective that is observable but is very challenging to mathematically formalize or quantify. Therefore, we argue that it should be considered during the design of the user simulator before assessing the proposed objectives, and especially the evaluation objective. On the other hand, realism is a complex objective that may be decomposed into several lower-level objectives. In this work, we propose the formalization of one of these. Indeed, a user simulator sharing a highly similar behavior with real users is one indicator of realism. However, other objectives should be formalized in future work to ensure a comprehensive assessment, e.g., consistency of actions taken given a context [5] and ability to learn and forget information [1]. This also includes the development of associated methodologies and metrics to evaluate the fulfillment of these objectives. It is possible that some of these lower-level objectives may not be critical for training and/or evaluation, but still improve the overall realism of the simulation.

The focus of the current work has been on the objectives of user simulation in the context of conversational information access. However, the general concepts defined here could be adapted to the broader context of interactive information access. Indeed, a session can be represented by replacing the sequence of utterances by a sequence of interactions between an information access system and a user (such as querying, clicking, and liking/bookmarking documents), suggesting a potential generalization to interactive information systems.

# 6 CONCLUSION

In this work, we have presented a formal characterization of the distinct objectives for user simulation for training and evaluation purposes in the context of conversational information access. We have demonstrated empirically that optimization on the training objective—maximizing behavioral similarity to real users—does not necessarily imply improvement on the evaluation objective—accurately predicting the performance of the conversational agent in helping users accomplish some task. This highlights the need for distinct design considerations when developing user simulators for training vs. evaluating systems. Whether optimizing for one objective inherently benefits the other remains an open question, motivating further research.

A key contribution of this work is the establishment of a formal framework for quantifying user simulator performance on these objectives. This framework enables direct comparison between different simulators and provides a foundation for guiding the development of simulators tailored to specific purposes.

We identify two main directions for future work. First, we plan to conduct a more comprehensive empirical study where we evaluate the performance of multiple conversational agents that are trained using different user simulators. Second, we aim to refine the evaluation metrics to provide a more nuanced and comprehensive assessment of user simulator effectiveness.

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
