# OpenReview forum: "Towards a Formal Characterization of User Simulation Objectives in Conversational Information Access"
_ACM.org/SIGIR/ICTIR/2024/Conference — ICTIR 2024_

### Official Review · Reviewer_sCyr · 2024-05-15

**Rating:** 2
**Confidence:** 3

**Objective Part Of Review:**

Brief Summary: The contributions of this paper are two-fold. First, user simulations in system evaluation are formalized and characterized using different objectives. Second, empirical experiments show that for user simulation, optimizing for one training objective (e.g., maximizing similarity to real users) does not necessarily correspond to improved performance on the evaluation objective.

- Soundness: The datasets, methods, experiments, and metrics are described clearly.
- Presentation: The paper is clear and easy to read.
- Difficulty: The paper’s contributions are highlighted in both their formalization of the problem and empirical results.

**Subjective Part Of Review:**

This paper provides nice formalizations for its methods. With respect to the empirical results, some of the metrics are grounded to specific characteristics (e.g., patience, inclination), but the paper provides a good starting point into further studies for discrepancies between user simulations objectives at train vs test time.

A few questions/comments:
- The number of dialogues used for each of the five user simulators vary drastically. Was the entire dataset used for each simulator?

---

### Official Review · Reviewer_xZ3T · 2024-05-16

**Rating:** 0
**Confidence:** 1

**Objective Part Of Review:**

The contribution is well written and describes a formal approach to the simulation of user behaviour during dialogue. The approach and the formalisation of the model seem technically robust. The experimental component is based on a very limited number of agents and users, making the results weak. Overall, the article seems convincing, although it is not clear to me the possible use of similar systems as they do not generate interactions that are verisimilar to the highly variable and unpredictable interactions of humans.

**Subjective Part Of Review:**

Unfortunately, the article is far removed from my research interests. This certainly supports my scepticism about the real usefulness of such approaches. Nevertheless, I do not feel able to provide a critical review and will support what more experienced reviewers than me will provide as suggestions.

---

### Official Review · Reviewer_89Ad · 2024-05-17

**Rating:** 1
**Confidence:** 4

**Objective Part Of Review:**

Overall, this paper provides a comprehensive analysis of the problem statement, methodology, results. The problem statement is clearly articulated, offering a clear and insightful, high level, presentation of the core content of the paper, before delving into more details. The authors effectively establish the research question and provide a structured approach to investigating it. The methods employed are well-described, offering clearly defined and utilized formal notations and concepts. Insightful elaboration is conducted regarding the distinction of the twofold objectives when training and evaluating user simulation systems, while also, regarding possible measures to evaluate such objectives. Moreover, a clearly stated empirical study is elaborated, addressing, showcasing that the two proposed objectives are not necessarily aligned. Results are presented in a systematic manner and all the claims are properly supported. The authors propose a series of metrics to evaluate user simulator performance and offer insightful discussions on the implications of their findings. No contradictions were spotted.

**Subjective Part Of Review:**

The paper offers a straightforward reading experience, presenting complex concepts in a clear and concise manner. The chosen chapter layout enhances readability, while the formal notations in use are adequately explained, making it easy for readers to follow the arguments of this study. The problem addressed is relevant, addressing an important subject for the growing trend of user simulation. Moreover, while some components of the elaborated methods draw from established methodologies or metrics, these are properly integrated in the current method. The results presented are interesting, shedding light on the complexities of user simulation objectives and their implications for system development. The observation that optimization for one objective does not necessarily lead to improvement on the other objective provides valuable insights for researchers and practitioners alike, prompting further exploration. As such, it is highly likely that others in the ICTIR community will find this work compelling. Only minor issues were identified. Firstly, the 2.1 subsection could be rewritten in a more concise way (e.g. The section discussing the "Simulated Agenda Dataset" lacks integration and coherence within the broader context. Improved organization and a clearer articulation of ideas through paragraph structuring would enhance readability and narrative cohesion. Refinement is necessary to ensure a seamless flow of concepts and topics.). Besides that, it would also be useful to provide more specific details in Chapter 4, regarding the interaction QRFA model in use, to facilitate the understanding of the reader.

---

### Meta-Review · Area_Chair_mUjx · 2024-05-30

**Recommendation:** Accept (Oral)
**Confidence:** 4

**Metareview:**

All reviewers agree that this paper offers a significant contribution to the analysis of user simulators in information accessing. The paper is well written and easy to follow, and the presentation of systematic results that are well-supported by the empirical experiments.

Though some reviewers have pointed out that the paper could be improved if more details are provided on models and applications, the overall merits of this paper clearly overwhelm its weakness. We recommend an accept and suggest the authors to revise the paper based on the review comments before submitting the camera-ready version.